# Connected, Respected and Contributing to Their World: The Case of Sexual Minority and Non-Minority Young People in Ireland

**DOI:** 10.3390/ijerph18031118

**Published:** 2021-01-27

**Authors:** András Költő, Aoife Gavin, Elena Vaughan, Colette Kelly, Michal Molcho, Saoirse Nic Gabhainn

**Affiliations:** 1Health Promotion Research Centre, National University of Ireland Galway, H91 TK33 Galway, Ireland; aoife.gavin@nuigalway.ie (A.G.); elena.vaughan@nuigalway.ie (E.V.); colette.kelly@nuigalway.ie (C.K.); saoirse.nicgabhainn@nuigalway.ie (S.N.G.); 2School of Education, National University of Ireland Galway, H91 TK33 Galway, Ireland; michal.molcho@nuigalway.ie

**Keywords:** adolescent health, psycho-social determinants of health, better outcomes brighter futures framework, BOBF, health behaviour in school-aged children study, HBSC, sexual minority youth, discrimination

## Abstract

Outcome 5 of the Irish Better Outcomes, Brighter Futures national youth policy framework (“Connected, respected, and contributing to their world”) offers a suitable way to study psychosocial determinants of adolescent health. The present study (1) provides nationally representative data on how 15- to 17-year-olds score on these indicators; (2) compares sexual minority (same- and both-gender attracted youth) with their non-minority peers. We analyzed data from 3354 young people (aged 15.78 ± 0.78 years) participating in the Health Behaviour in School-aged Children (HBSC) study in Ireland. Age and social class were associated with the indicators only to a small extent, but girls were more likely than boys to report discrimination based on gender and age. Frequency of positive answers ranged from 67% (feeling comfortable with friends) to 12% (being involved in volunteer work). Sexual minority youth were more likely to feel discriminated based on sexual orientation, age, and gender. Both-gender attracted youth were less likely than the other groups to report positive outcomes. Same-gender attracted youth were twice as likely as non-minority youth to volunteer. The results indicate the importance of a comprehensive approach to psycho-social factors in youth health, and the need for inclusivity of sexual minority (especially bisexual) youth.

## 1. Introduction

### 1.1. The Human Ecology of Child and Adolescent Health

Health is situated within a complex, interactive network of determinants [1]. These determinants can be understood as the personal, social, economic, and political conditions in which people grow, live, work, and age, and the structural drivers behind those conditions [2]. Research, as well as practice, often concentrates on intra-individual factors at the expense of social determinants, despite the need to also address these social factors if we want to achieve structural equality in health [3]. Bronfenbrenner’s ecological systems theory, originally developed to better understand human development and children’s health [4], provides a useful framework to understand how these complex factors shape health outcomes across individual, micro-, meso- exo-, and macro-levels over the life-course of an individual [5].

Currently, we are seeing a shift in focus from negative to positive determinants and outcomes in child and adolescent health research [6]. These include concepts such as resilience, self-esteem, sense of connectedness, and identity development. Such factors are embedded within the microsystems, including young persons’ family and peer networks and their interactions with each other and with extended family and neighborhoods, schools, media, and other venues of social life (exosystem). Intra-individual factors (e.g., age and gender) and macro-level impacts such as socio-economic status and political environment are also salient. These are embedded within the life course of the young people as well as the conditions and events of the historical period in which they live. In Bronfenbrenner’s model, these two effects constitute the chronosystem. An increasing corpus of evidence shows that positive psycho-social determinants such as connectedness (family, peer, and school support) [7,8,9], self-esteem [10,11], volunteer work and community engagement [12], or perceived freedom [13] are related to better health outcomes.

Despite this shift to a more positive approach, it must be acknowledged that many young people who belong to marginalized groups [14], including youth from immigrant families, youth living in poverty, those who have a disability or chronic condition, young carers, Travellers (an indigenous ethnic minority group in Ireland), and those who identify as lesbian, gay, bisexual, transgender, intersex, or belonging to another sexual or gender minority (LGBTI+), often experience discrimination. 

Young people with multiple, intersecting minority statuses may face even more discrimination and associated health disparities. Actual, as well as anticipated or perceived, discrimination [15] may negatively impact mental and physical health through the moderating effect of heightened stress responses or health risk behaviors (e.g., alcohol consumption or smoking), which are potential coping mechanisms when an individual experiences discrimination [16]. Due to developmental social processes, adolescents are especially vulnerable to discrimination based on many grounds, including their or their families’ ethnicity, their gender, age, physical characteristics such as weight and height [17], or sexual orientation [16,18]. Experiences of discrimination in adolescents have been associated with decreased self-esteem and increased depressive symptoms [19], and there is evidence that their negative impact on health can last long into adulthood [20].

### 1.2. Connected, Respected, and Contributing to Their World: The Fifth Outcome of the Better Outcomes, Brighter Futures

In Ireland, the health and well-being of youth are prioritized in line with the State’s commitments under the United Nations Convention on the Rights of the Child. This was the basis of the development of Better Outcomes, Brighter Futures (BOBF), the national policy framework that sets out the Government’s agenda and priorities in relation to children and young people up to the age of 24 years [21]. The BOBF indicator set [22] was developed to track progress for children and young people across five national outcomes for the purposes of identifying trends, contributing towards priority setting, and to inform policy and service provision. The five dimensions (“outcomes”) of BOBF state that children and young people will (1) be active and healthy; (2) achieve in all areas of learning and development; (3) be safe and protected from harm; (4) experience economic security and opportunity; and (5) be connected, respected, and contribute to their world. The indicators reflect a broad picture of children and young people’s lives and their experiences. These outcomes are examined against an indicator set that was completed following a comprehensive development and consultation process. The BOBF indicator set includes more than 100 indicators across 70 indicator areas. A national research priority is that country-level population health studies provide data to these indicators.

One of the main data providers of BOBF is the Health Behaviour in School-aged Children (HBSC) Ireland study. HBSC is a World Health Organization collaborative cross-national study initiated in 1983 and thence conducted in four-year cycles, in a constantly growing number of countries. Currently, 51 countries in North America, Europe, and the former Soviet region participate in the study [23]. The Irish HBSC asks children and young people aged 10–17 about their health and well-being and health behaviors within their social contexts—home, school, and with family and friends. It is a school-based survey, with data collected through self-completion questionnaires administered by teachers in the classroom. There is a strong conceptual and methodological connection between the HBSC study and the BOBF framework. This is illustrated by the fact that in the 2018 survey round, the HBSC Ireland study was the data source for more than one third (31) of the BOBF indicators.

The fifth outcome of BOBF—“Connected, respected, and contributing to their world”—contains four aims (Table 1): Young people (1) develop a sense of identity and are free from discrimination; (2) are part of positive network of friends, family, and community; (3) are civically engaged, socially and environmentally conscious; and (4) are aware of their rights and are responsible and respectful of the law. These aims are assessed with thirteen indicator areas, and HBSC Ireland provides data on ten of these (Table 1). The aims, indicator areas, and the actual indicators collected by HBSC are outlined in the Method section. There are some areas within BOBF Outcome 5 where other studies provide data to the indicator set. The Quarterly National Household Survey, Special Module on Equality, conducted by the Central Statistics Office of Ireland provides data on perceived discrimination in 18- to 24-year-olds [24]. The Growing Up in Ireland longitudinal study provides data on peer support and having at least one caring and consistent adult to confide in among 9-year-olds [25]. The Programme for International Student Assessment (PISA) provides data on positive parent and family relationships [26]. However, the HBSC indicators relevant to BOBF Outcome 5 cover the majority of the indicator areas and therefore provide an extensive picture of adolescents’ connectedness, sense of being respected, and contribution to their world. The first aim of the present article is to demonstrate how population-level descriptive data on BOBF Outcome 5 (“Connected, respected, and contributing to their world”) can inform the implementation of a national level policy to improve adolescents’ lives. We hypothesize that age, gender, and socio-economic status will impact some of the BOBF indicators among 15- to 17-year-old youth in Ireland.

### 1.3. Connectedness, Self-Esteem, and Societal Engagement in Sexual Minority Youth

An increasing number of scholars studying sexual (and gender) minority youth health acknowledge that the dominant narratives describing lesbian, gay, and bisexual (LGB) youth as universally vulnerable should be shifted to focus on the positive aspects of queer identities and lives [27,28,29].

Such a positive, “after-queer” [30] approach does not mean neglecting or downplaying that many sexual and gender minority youth face challenges and adversity, including experiences of stigma and discrimination [31]. It rather emphasizes that a disproportionate focus on negative aspects of sexual and gender minority lives may overshadow or impede the benefits and positive aspects of LGBT identities. Riggle and Rostosky [32] discuss many positive LGB (and transgender) experiences uncovered by their research, including living an authentic life, having empathy and compassion for others, belonging to a community, and experiencing strong emotional connections with others. These latter themes of belonging and connectedness were also highlighted by a recent systematic review of qualitative research on sexual and gender minority youth [33]. The authors found that connectedness with others, whether as part of small groups like Gay–Straight or Gender–Sexuality Alliances, wider networks of sexual and gender minority youth, or online friendships, can be a source of empowerment as well as a vital protective factor. However, evidence on such positive factors which promote resilience are scarce, the findings are inconsistent, and—just like with many other facets of LGB research—the majority of studies were conducted in North America [34,35].

While there have been some studies with sexual minority youth in Ireland [36,37,38], the research landscape is rather bleak, and the existing studies concentrate on the negative aspects of belonging to sexual minorities. The *LGBTI+ National Youth Strategy 2018–2020* [39] prioritizes conducting more comprehensive and balanced research to better understand the specific needs, and developmental assets, of sexual and gender minority youth. In recent years, Ireland has advanced LGBTI+ equality at a structural level, with the passing of marriage equality legislation and the Gender Recognition Act in 2015. However, there is still much work to do in achieving more inclusive environments for LGBTI+ people. For instance, LGBTI+ young people still report a lack of understanding and acceptance [40], as well as barriers to accessing inclusive and supportive services [41].

Stigmatization and discrimination are contingent on social and cultural context and occur across different levels of the socio-ecological spectrum [44]. Sexual minority youth who experience higher levels of stigma may be at increased risk of adverse health outcomes, which can vary depending on the mechanism of stigma experienced [31,45]. LGB young people can also experience compound or intersecting stigmas, related to other aspects of their identity, including chronic conditions and disability status, race, ethnicity, faith, sex, and gender identity. For instance, a meta-analysis found a stronger correlation between minority stress and negative mental health outcomes in lesbian and bisexual females than in gay and bisexual males [46]. These findings highlight the need to adopt a more nuanced approach that considers the heterogeneity of young people’s lives. Such an approach should account for the wide variety of context-specific, environmental, social, and structural factors involved in shaping LGBT youth health and well-being and balance the focus to involve positive determinants of health.

One of the potential ways to utilize the BOBF indicator set is to explore whether perceived discrimination affects the sense of being respected and valued and community engagement. Comparing sexual minority and non-minority youth on these dimensions speaks to the *LGBTI+ National Youth Strategy 2018–2020* of Ireland [39], the first governmental strategy in the world specifically aimed to improve the health and well-being of sexual and gender minority young people. This strategy is directly aligned with the BOBF framework but sets specific objectives to improve the health and lives of sexual and gender minority youth in Ireland. An additional benefit of applying the BOBF framework in the study of sexual minority young people is that the findings may add to the thin evidence basis on connectedness and self-esteem in LGB youth.

Therefore, the second aim of our paper is to compare sexual minority and non-minority youth in Ireland across the indicators included in BOBF Outcome 5. We hypothesize that compared to their non-minority peers, sexual minority youth will be more likely to feel discriminated based on their sexual orientation and other grounds; that they will be less likely to report high levels of social support (from families and peers); and that they will be more likely to be involved in volunteer work.

## 2. Method

### 2.1. Procedure

HBSC is a cross-sectional epidemiological study conducted every four years with nationally representative samples of children and adolescents. In line with the international HBSC protocol [42], a two-stage cluster sampling was conducted to ensure national representativity. First, a proportional sample of schools was randomly selected from the eight geographical regions of Ireland; subsequently, class groups within the participating schools were also randomly selected [47]. School principals were contacted by post; 63% of the invited schools agreed to participate. Data collection took place between April and September 2018. In the present study, responses from an average of 11 young people from 297 classes were analyzed. The study instrument was a paper-based questionnaire that participating youth filled in during school hours. The study was carried out in adherence to the international HBSC study protocol [42] and was approved by the Research Ethics Committee of the National University of Ireland Galway under Decision Ref. REC17-Nov-13. Informed consent was obtained from all participating youth as well as their parents/guardians and school principals. It was at the discretion of the school principals and boards whether active or passive consent from parents was required. Evidence shows that reported health outcomes in young people are independent from the type of parental consent [48], therefore, we have not used it as a control variable. No reimbursement was offered or provided for participation. Young people were told that they were free to not answer any questions in the survey and to withdraw their participation at any time.

### 2.2. Measures

#### 2.2.1. Sociodemographic Variables

*Gender* was assessed by a single item: “Are you a boy or a girl?”, with response options “A boy”/“A girl”. Age was computed from the year and month of birth reported by the respondents, in combination with the time of data collection. *Social class* was manually coded based on the parents’ occupation and employment reported by young people, into the categories used by the Central Statistics Office [49]. The highest social classes included children of professional and managerial or technical workers. Medium social class groups comprised children of non-manual and skilled manual workers. The lowest social class groups included children of semi-skilled and unskilled workers.

#### 2.2.2. Romantic Attraction

A single item was used to assess *romantic attraction* in young people, developed by members of the sexual health working group of the international HBSC Network [50]: “Are you attracted to…”, with response options “Girls”/“Boys”/Both girls and boys/“I am not attracted yet to anyone”. Boys attracted to girls and girls attracted to boys were coded as attracted to opposite-gender partners. Boys attracted to boys and girls attracted to girls were classified as attracted to same-gender partners. Those indicating being attracted to both girls and boys were classified as attracted to both gender partners. Those reporting not being attracted to anyone were classified as not attracted.

#### 2.2.3. BOBF Outcome Indicators

*Perceived discrimination* was assessed by a grid of items (Table 1): “How often are you treated unfairly or negatively…”, based on the eight grounds listed by the Equal Status Act 2000 [51]. The items included unfair or negative treatment because of where the respondent, their parents, or grandparents were born; because of the respondents’ gender; their age; their disability; their race; their sexual orientation; their religion; or their membership of the Traveller community. The frequency of these were rated by the respondents on a five-point Likert scale, with response options “Never” = 1, “Hardly ever” = 2, “Sometimes” = 3, “Often” = 4, and “Very often” = 5. The numeric responses were used as continuous outcome variables. A ninth item, discrimination based on other reason(s), was also administered. Alongside the frequency, respondents could provide reason(s) in a text box. Perceived discrimination based on other ground(s) is not included in the present analysis.

Single items were used to cover the following indicator areas (Table 1). *Experience of sense of freedom*: “In general, do you feel you have freedom in your life?” *Having at least one caring and consistent adult to confide in:* “In general, do you have a caring adult you can tell anything to?” *Perceptions of being valued and respected:* “In general, do you feel valued and respected?” *Belief in being able to make a positive contribution to the world:* “In general, do you feel that you make a positive contribution to the world?” *Volunteering and altruism:* “In general, do you take part in volunteer work?” *Children and young people’s awareness of their rights:* “In general, do you know your rights as a young person?” For these seven areas, the response options were: “1 (Not at all)”/“2”/“3”/“4 (Very much)”. The response options were dichotomized so that 1 reflects very much, while 0 reflects responses less than very much.

The eight discrimination items described above and six single items, developed by the HBSC Ireland team specifically to cover BOBF Outcome 5, were tested in a sample of 237 young people [52]. This pilot phase included asking young people to mark on their questionnaire words or phrases within the questions or response options they did not understand. Following completion of the questionnaire, a classroom discussion was facilitated to ensure that the items were understandable and acceptable. Apart from one respondent who asked what “race” is, and another asking what “sexual orientation” is, young people reported no problems with the content or wording of any item [52].

For *Peer acceptance and respect*, we used an item developed by children in Ireland as part of a youth consultation process [53]: “Do you feel comfortable being yourself while with your friends?” Response options were: “Always”/“Often”/“Sometimes”/“Never”. These were dichotomized into 1 (always) and 0 (less often than always) (Table 1).

*Positive parent and family relationships* were operationalized by the Family subscale of the Multidimensional Scale of Perceived Social Support (MSPSS) [43]. This subscale contains four items which capture helpfulness and emotional availability of the family. Respondents rated all items on a seven-point Likert scale where 1 indicates very strong disagreement and 7 very strong agreement. In line with the international HBSC protocol [42], scores for the four items are summed and divided by four, and the overall score is dichotomized in a way that a score 5.5 or above indicates high family support, while a score lower than 5.5 indicates low family support. In our sample, the scale had high internal consistency (Cronbach’s alpha = 0.956). *Positive relationships with peers* were assessed by the Friends subscale of the MSPSS [43]. Similar to the Family subscale, it contains four items on friends’ helpfulness and emotional availability. Response options and scoring are identical to that of the Family subscale, thus, based on the cut-off score of 5.5 or above, we coded high friend support, while a score of less than 5.5 was coded as low friend support [42]. The items showed high internal consistency (Cronbach’s alpha = 0.959).

Table 1 shows three BOBF indicator areas where HBSC is not providing data. The area *18–24-year-olds who vote in local, regional, national, or European elections and referenda* concerns young people outside the age range of the HBSC Ireland study. *Respect for laws and the judicial process* and *Perception of fairness of the law* fall outside the scope of HBSC. Therefore, these three indicator areas are not featured in the present analysis.

### 2.3. Sample

The present study includes 15- to 17-year-old young people attending post-primary schools in Ireland. The selection of the sample is displayed in Figure 1. From the full sample, we eliminated those who were younger than 15 years, or did not provide information on their gender, age, social class, or romantic attraction. At this stage, the sample comprised data from 3354 young people, with a mean age of 15.78 (SD = 0.78), with 55.1% girls; 56% belonging to the highest social classes, 34.3% to the medium social classes, and 9.7% to the lowest social classes. Attraction-wise, 88.4% reported being exclusively attracted to opposite-gender partners, 3.0% exclusively to same-gender partners, 6.3% to both gender partners, and 2.4% reported no attraction.

Subsequently, all those who provided answers to the BOBF Outcome 5 indicators were retained for the analyses (3190 ≤ *n* ≤ 3342). Since the responses on discrimination items were analyzed together, those who did not answer one or more items on discrimination had to be excluded, resulting in *n* = 3213 for the discrimination analyses.

### 2.4. Analytic Approach

Analyses were carried out in SPSS 24.0 (IBM Corp., Armonk, NY, USA). Since the variables on perceived discrimination were continuous, while all other items were dichotomized, we used two arms of statistical analysis for each study aim.

For Aim 1 (presenting population-level descriptive statistics), first the means and standard deviations are provided by gender and social class groups. Discrimination perceived by boys and girls was compared by Student’s *t*-tests. Perceived discrimination across social groups was compared by one-way variance of analysis (ANOVA). Effect size *r* for *t*-tests and omega-squared effect sizes for ANOVA were calculated. For all other BOBF Outcome 5 variables, the percentages of positive responses are provided. The association of the variables with gender and social class were examined by chi-square tests. Cramér’s *V* effect sizes are provided for these associations.

For Aim 2 (analysis of the outcomes across romantic attraction groups), first means and standard errors of perceived discrimination are provided for the four attraction groups. Data on discrimination were heavily skewed (the majority of young people reported never or hardly ever being discriminated on any grounds). According to Kolmogorov–Smirnov tests, all of these variables deviated from normal distribution (*p* < 0.001). Therefore, medians for each grounds of discrimination across romantic attraction groups were obtained. To the same end, the means were compared using non-parametric (Kruskal–Wallis test) as well as parametric methods (ANOVA). For the latter, indices of omega-squared effect size and statistical power are also provided. The magnitudes of the effect sizes were interpreted following Cohen’s (1988) guidelines [54]. We compared means pairwise across romantic attraction groups, adjusted for Šidák criteria. To make interpretation easier, mean values of perceived discrimination across romantic attraction groups are displayed in a vertical profile chart (Figure 2), where the *y* axis represents grounds of discrimination, and the *x* axis indicates mean frequency of perceived discrimination. The vertical lines in the chart represent the pattern of discrimination experienced by the different groups. A similar profiling approach is used in studies of personality [55]: an example of such vertical profile charts is provided by Carver and Scheier ([56], p. 38).

For all other BOBF Outcome 5 variables, binary logistic regression models were computed, with opposite-gender attracted young people as the reference group. Significances of the models were tested by Wald chi-square tests. Odds ratios were obtained to assess whether same-gender attracted, both-gender attracted, and not attracted young people had different outcomes than their opposite-gender attracted peers. For all odds ratios, 95% confidence intervals (CIs) were calculated. Statistical significance for all analyses was set at *p* < 0.05.

At different stages of the analysis, we conducted statistical tests adjusted for gender, social class and other variables, including area of residence, parental immigrant status, young people’s disability, or chronic conditions, and Traveller ethnicity. Ultimately, we decided not to adjust the analyses, for two reasons. First, their effect was usually not significant, or even if significant, the effect sizes were marginal and/or lacking adequate statistical power. Second, including these as control variables or covariates poses methodological and theoretical problems, which will be outlined in the Discussion.

## 3. Results

### 3.1. Descriptive Findings: Discrimination

Mean scores for frequency of perceived discrimination, along with standard deviations, are displayed in Table 2. Means are given for boys and girls, and the three social class groups separately. Boys and girls reported similar levels of discrimination based on their disability, race, sexual orientation, and religion. Boys were significantly more likely to report being discriminated based on their, their parents’ or grandparents’ place of birth, or belonging to the Traveller community, but the effect sizes were low. Girls were significantly more likely to be discriminated based on their age and gender. The effects were medium sized. Across social classes, no significant difference was found in perceived discrimination based on young people’s gender, age, race, or religion. Young people from lower social classes were more likely to report discrimination based on their (or their parents’) birthplace or their Traveller status, but the effect sizes were marginal. Young people from medium social classes reported somewhat more frequent discrimination based on their sexual orientation than their peers in high or low social classes, but this effect was also marginal. The means rarely exceeded 2, which indicates that the majority of young people never or hardly ever experienced discrimination based on any of the eight grounds.

### 3.2. Descriptive Findings: Positive BOBF Outcome 5 Variables

Percentages of young people giving a positive answer to the positive BOBF Outcome 5 indicators (e.g., reporting that they feel freedom in their lives, always feeling comfortable being themselves while they are with their friends, very much agreeing with having a caring adult, etc.), are presented in the upper part of Table 3. These showed a large variation: more than two thirds of young people reported always feeling comfortable being themselves while they are with their friends and almost 60% reported high level of peer support, while only around a fifth indicated feeling that they made a positive contribution, and around one in ten reported that they took part in volunteer work. Perceived freedom, feeling comfortable while being with friends, high family support, feeling valued and respected, and being aware of one’s rights as a young person were not associated with gender. Girls were significantly more likely to report having a caring adult and taking part in volunteer work, and boys were more likely to feel that they made a positive contribution than girls, but the effects were small. Girls were significantly more likely to report high peer support; this effect was medium in size. In general, the BOBF Outcome 5 indicators were not associated with social class. However, young people in medium social class groups were significantly more likely to feel comfortable with their friends, and those in high social classes more likely to report high family support compared to their peers in other social class groups (*p* ≤ 0.004); the effects were small to medium in size.

### 3.3. Comparing Sexual Minority and Non-Minority Youth: Discrimination

Numerical data and comparisons of perceived discrimination across romantic attraction groups is presented in Table 4. The mean values for discrimination are also displayed in a vertical profile chart (Figure 2), where the romantic attraction groups are marked with separate lines. The means and the medians indicate that most young people, regardless of their romantic attraction, never or only rarely experienced discrimination based on any grounds. Both parametric ANOVA and non-parametric Kruskal–Wallis tests rendered significant results in almost all cases, indicating that there are significant differences in perceived discrimination across romantic attraction groups. Nevertheless, the models had marginal effect size for discrimination based on birthplace, disability, race, and religion. Discrimination across romantic attraction based on belonging to the Traveller community was not significant when non-parametric comparison was used, which may reflect the small proportion of Traveller youth in our sample, and that only a fraction of them reported being attracted to same-gender or both gender partners. This is supported by the model being on the border of acceptable statistical power (0.827).

There were some grounds, though, on which same- and especially both-gender attracted young people were more likely to feel discriminated than their opposite-gender attracted and not attracted peers. Significant differences were detected across romantic attraction groups for gender and age, with small effect sizes. Pairwise comparisons revealed that both-gender attracted youth were significantly more likely than the other three groups to feel that they were discriminated based on their gender and age. In other cases, apart from belonging to the Traveller community, both-gender attracted young people also experienced more discrimination than the other groups. Discrimination based on sexual orientation was significantly more likely to be experienced by same- and both-gender attracted young people than their opposite-gender attracted and not attracted peers. The effect size was large.

### 3.4. Comparing Sexual Minority and Non-Minority Youth: Positive BOBF Outcome 5 Variables

The lower part of Table 3 presents the proportion of positive responses to BOBF indicators other than discrimination. Perceived freedom, feeling that one made a positive contribution, and knowing their rights as a young person were not significantly associated with romantic attraction. Feeling comfortable while being with friends, having a caring adult, high family and peer support, feeling valued and respected, and taking part in volunteering work were significantly associated with romantic attraction, but the effect sizes were small.

This pattern was reflected in the binary logistic regression models (Table 5): romantic attraction, overall, did not have a significant effect on perceived freedom, feeling that they made a positive contribution, and knowing their rights as a young person. In other words, same- and both-gender attracted and not attracted young people had odds similar to their opposite-gender attracted peers of reporting these positive outcomes. It should be noted, though, that both-gender attracted youth were 0.7 times less likely, compared to their opposite-gender attracted peers, to feel that they made a positive contribution; while same-gender attracted youth were 1.3 times more likely to report that they knew their rights as a young person.

Both-gender attracted young people were, significantly, 0.8 times less likely to report having a caring adult compared to their opposite-gender attracted peers; similarly, they were less likely to report high family support and feel that they made a positive contribution. Same-gender attracted and not attracted youths’ odds were not significantly different from that of their opposite-gender attracted peers. On the other hand, both-gender attracted and not attracted youth had significantly lower odds of reporting feeling comfortable while being with their friends compared to those who reported opposite-gender attraction. No significant difference was found for the odds of same-gender attracted youth. Not attracted young people were also less likely than the other three groups to report high peer support.

Same-gender attracted young people were almost twice as likely as their opposite-gender peers to report taking part in volunteer work. Both-gender attracted youth were also somewhat more likely to report volunteering, but this was not statistically significant.

## 4. Discussion

In this study, we analyzed the prevalence of psycho-social phenomena under the Better Outcomes, Brighter Futures (BOBF) national youth policy framework’s fifth outcome (“Connected, respected, and contributing to their world”) in a nationally representative sample of 15- to 17-year-old young people in Ireland, demonstrating the utility of population-based data in informing policy makers. Data for the majority of the BOBF Outcome 5 indicators are provided by the Irish Health Behaviour in School-aged Children (HBSC) study, a World Health Organization collaborative cross-cultural study, therefore, we examined a subsample of the HBSC Ireland study, conducted in 2018. The first aim of the present study was to provide population-level data on these outcomes. The second aim was to showcase the utility of the BOBF framework by comparing how sexual minority (same- and both-gender attracted) young people fare on these indicators compared to their non-minority (opposite-gender attracted and not attracted) peers.

### 4.1. Descriptive Findings: Gender and Social Class Have a Moderate Role

In relation to Aim 1, a positive finding of the study is that a large majority of young people were not likely to report that they experienced discrimination based on any of the eight grounds listed by the Equal Status Act of Ireland [51]: these include where the respondent, their parents, or grandparents were born; the respondents’ gender; their age; their disability status; their race; their sexual orientation; their religion; or their membership of the Traveller community. In general, perceived discrimination was not associated with gender or social class. Even where associations were statistically significant, the sizes of the effect were small. In that sense, our hypothesis outlined in Section 1.2 was not confirmed by the data. Remarkable exceptions were that girls were more likely to feel that they had been discriminated based on their gender or their age, with medium effect sizes. While there is little consensus in how to characterize and analyze intersectional stigma [57], this result suggests that one component that contributes to intersecting discrimination is being female. Ageism—prejudices based on somebody’s biological age—is often experienced by young people; this is often combined with sexism at the expense of girls or young women, especially if they are involved in activism or act in a non-traditional manner [58]. On the other hand, girls were significantly more likely to report high levels of peer support than boys. This is in line with earlier findings, i.e., that girls spend more time with their friends and perceive their friendships as more cohesive and emotionally close than boys [59]. These results indirectly suggest that interpersonal solidarity among minority girls may, to some extent, counterbalance the negative effects (e.g., stress and anxiety) stemming from discrimination [60]. This may be an important asset in interventions that aim to improve the lives of sexual minority girls.

Only 11.5% of the young people agreed very much with that they are participating in volunteer work. In a seven-country study, adolescents reporting civic commitments, regardless of their gender or country, were more likely to consider public interest an important life goal [61]. Percentages of young people reporting ever being engaged in volunteering work ranged from 16% (boys in Russia) to 68% (girls in Hungary), but they indicated their volunteering activity on a binary “yes-or-no” question, while we have used more stringent dichotomization. Had we included all youth who reported doing *any* amount of volunteer work, the percentages would have been 60.2% in boys and 63.9% in girls. Existing evidence unequivocally supports that young people who volunteer show more favorable outcomes in a large range of health, well-being, and psycho-social indicators than their peers who are not engaged in volunteering [62]. However, research on prosocial behaviors such as volunteering in adolescents is scarce, and the causal links between positive health, psycho-social characteristics, and prosocial behaviors are not fully understood [63]. A potential explanation is that engaging in volunteering can increase a person’s social capital, which is known to have benefits for health outcomes [64]. Another mechanism may be that being connected to communities can improve confidence and self-esteem in young people [65]. A key element of psychotherapeutic interventions for gay and bisexual men is to support them in considering how local LGBTI+ communities may help them in reducing stress and anxiety [66]. Facilitating joining such communities and/or being engaged in volunteering may be particularly beneficial for bisexual or both-gender attracted young people; however, it should be noted that a barrier for them belonging to LGBTI+ communities may be the anticipation that they will be discriminated against based on their bisexuality [67]. More explorative studies are required to better understand the specific needs and experiences of bisexual youth in communities and with voluntary work.

### 4.2. Better Outcomes, Brighter Futures in Sexual Minority Adolescents: Discrimination, Resilience, and Social Agency

Our study’s second aim was to showcase the utility of the BOBF Outcome 5 indicator set by comparing young people belonging to sexual minorities (operationalized by being attracted to same- and both gender partners) to their non-minority peers (being attracted exclusively to opposite-gender partners or not reporting being attracted). Sexual minority groups were more likely to perceive discrimination based on their sexual orientation, which is in line with our second hypothesis. This finding indirectly supports that assessing romantic attraction may correctly identify young people whose self-defined sexual orientation would have been lesbian, gay, or bisexual [68]. Elsewhere, our team have argued that classifying sexual minority young people based on the gender of partners they are attracted to may result in a larger group than assessing self-defined sexual orientation, since there may be young people who have not recognized and/or disclosed their non-heterosexual orientation [50].

Sexual minority young people, especially those who reported being attracted to both boys and girls, were more likely than their peers (even more than exclusively same-gender attracted youth) to feel discriminated against based on their age and gender. This result is in line with findings from the Hungarian LGBT Survey conducted in 2007, where researchers observed that members of the lesbian, gay, bisexual, or transgender community sample, compared to a nationally representative sample, were more likely to experience discrimination based on not only on the grounds of their sexual orientation or gender identity, but on other grounds as well [69]. The most cited other grounds for discrimination, similar to our study, were gender and age. We hypothesize that this additional discrimination may be attributed to a “negative halo effect”, sometimes termed “reverse halo effect” or “horns effect”. This phenomenon denotes that one characteristic that is considered as negative sheds a negative light on the whole person [70]. Thus, it is antithetical to the “halo effect”, described 100 years ago by E. L. Thorndike, when a (positive) characteristic of a person favorably biases how others evaluate their (otherwise unrelated) attributes [71].

Perceived freedom, the feeling that one makes a positive contribution to the world and knowing their rights as a young person were not significantly associated with romantic attraction. However, the pattern of the data suggests that had the subsample sizes in sexual minority and not attracted youth been larger, we would probably have seen significant differences.

Not attracted young people were less likely than their opposite-gender attracted peers to feel comfortable with their friends or report high peer support. A probable explanation is that young people who have never been involved in romantic relationships in general seem to have problems with establishing and building friendships with their peers [72], although from this association we cannot infer causality.

Both-gender attracted young people were less likely than their non-minority peers to report high family support, feeling comfortable while being with friends, or to feel valued and respected. Remarkably, no such difference was found for exclusively same-gender attracted youth. This result echoes earlier findings that adolescents who identify as bisexual face disproportionate health risks across a wide range of determinants and health outcomes compared to their peers who identify as heterosexual, or even lesbian or gay [73,74,75]. Our international HBSC team found that among adolescents from eight European countries or regions, both-gender attracted youth were the most likely to report substance use [76] or rate their health as poor and report multiple health symptoms [77]. This pattern of unfavorable outcomes for both-gender attracted or bisexual youth may be because bisexuality is often “invisible” or denied by members of the individuals’ social network. Bisexual people may be mis-classified by others as either heterosexual or lesbian/gay. Sometimes bisexual individuals experience rejection and biphobic harassment or monosexism (the belief that hetero- and homosexuality are superior to bisexuality) not only from their heterosexual peers but even by those who identify as gay or lesbian [78,79]. Some findings indicate that bisexual identity and gender interact with each other in their impact on health outcomes [80].

While from our data we cannot infer the reason for these disparities, it is worth noting that similar to many other Western countries, there is a growing acceptance towards LGBT individuals and issues in Ireland [81]. However, it seems that bisexual people do not benefit from this shift as much as their lesbian and gay peers. While being lesbian or gay can be perceived by others as a stable identity, bisexuality is often seen as “just a phase” and bisexual individuals report frequent discrimination, identity invalidation, and erasure [82]. Therefore, not only individuals need support and empowerment in being more bi-inclusive, but LGBTI+ communities as well.

Youth exclusively attracted to same-gender partners were almost twice as likely as their opposite-gender attracted peers to report that they are involved in community work. A similar, albeit not significant, tendency was observed among both-gender attracted youth. The work of Riggle and Rostosky [32] demonstrates that sexual minority people, due to their numerous experiences of discrimination and rejection, may become more aware of social injustice and empathetic with marginalized groups—not only based on sexual orientation or gender identity, but on other grounds too. This may motivate them to develop compassion towards others who are oppressed and stigmatized, and combat injustice and marginalization. In a recent landscape and knowledge gap analysis to explore and synthesize research done with LGBTI+ young people in Europe [83], we have found some studies tangentially dealing with community work, volunteering, and activism. These imply that many sexual and gender minority young people may want to do volunteer work for their local LGBTI+ communities, which provides them with a sense of belonging, or gives an opportunity to help others [84,85,86,87].

Mayock [88] found that LGBT young people in Ireland saw their community as a source of resilience and support, with themes of connectedness, safety, and solidarity frequently mentioned in respondents’ accounts of belonging to the LGBT community. Our findings, however, suggest that this may not be the case for all subgroups within sexual minority youth. Findings elsewhere indicate that LGBT young people report that dealing with adversity can result in a sense of empowerment [89], sometimes conceptualized as “coming out growth” [90]. Such intra-individual processes, coupled with LGBT community connectedness at the micro- and meso-level [91] and societal acceptance and tolerance at the macro-level [81,92], may act to boost self-esteem, increase resilience, and buffer the negative effects of stigma and discrimination.

### 4.3. Limitations and Strengths

Both gender and romantic attraction questions provided only binary (boy–girl) response options which exclude gender diverse (transgender, non-conforming, or non-binary) young people, or those living with an intersex variation. Romantic attraction is only one dimension of sexual orientation beside sexual identity, behavior, and fantasies, and there may be young people who would have described their gender in other terms than “boy” or “girl”. While dimensions of sexual orientation are generally overlapping [68], an inclusive assessment of sexual orientation needs to capture them. The BOBF is a comprehensive framework to improve the health and well-being of all young people containing a large suite of indicators, but it inevitably misses some key aspects of sexual and gender minority youths’ lives (e.g., coming out or disclosing someone’s sexual orientation or gender identity) to different persons. We have not assessed adolescents’ romantic relationship status either. Other key aspects to provide a balanced, non-victimizing account of sexual minority youths’ life and well-being would require that we assess resilience in the face of sexual orientation- and gender identity-based harassment, exclusion, and bullying, as well as belonging to LGBT communities. Finally, it should be noted that since HBSC collects data in classrooms, young people who were absent on the day of the data collection or attend youth centers or out of school services will inevitably be excluded from the sample. Given that sexual minority youth tend to miss school due to health problems [77] or due to fear of harassment and bullying [36], they are probably underrepresented in the present study. Further work is needed to include a broader sample of young people both within and outside the traditional school setting (e.g., by using community sampling).

As noted in the Method section, adjusting for “background” variables caused a dilemma. In earlier stages of the analysis, we controlled the comparisons of romantic attraction groups for gender and social class, but either these were not significant predictors of the outcomes, or—due to the very low subsample sizes—they resulted in inadequate statistical power. However, there is the broader problem of intersectionality [93]. It has to be noted that the results, especially those on perceived discrimination, are confounded with sexual minority and non-minority young people potentially belonging to other minority or marginalized groups, for instance, based on their ethnicity, place of origin, belonging to the Traveller community, being a young carer, or living with a disability or chronic condition. Excluding them would have rendered much smaller subsamples, inevitably reducing statistical power. Potential solutions for these problems are case–control matching and identifying young people with multiple or intersecting minority statuses. Our team has previously used such analytic techniques, but these are not without methodological challenges [14].

On the other hand, we believe that our study has some strengths. It is based on a nationally representative sample of 15- to 17-year-olds from an international study using rigorous methodology. Micro- and meso-level psycho-social determinants of young people’s health were analyzed using an established, comprehensive indicator set. Some areas explored in this study (e.g., perceived discrimination, feeling valued and respected, and volunteering) are relatively less studied, especially among sexual minority adolescents.

### 4.4. Implications for Practice, Policy, and Research

In recent years, Ireland has seen substantial change in acceptance and tolerance towards LGBTI+ individuals, with the Marriage Equality Bill in 2015 allowing same-sex couples equal rights in marriage passed in a referendum by a popular vote of 62%. This demonstrated a remarkable shift in the attitudes towards the LGBTI+ community in a country where the constitution and societal values are historically grounded in the Catholic faith. However, this change in itself does not necessarily mean that sexual (and gender) minorities are now free from stigma and discrimination. Our findings, in line with earlier studies conducted in Ireland, found that sexual minority youth, especially both-gender attracted or bisexual young people, are more likely than their non-minority peers to experience discrimination and less likely to score favorably on psycho-social variables. It is therefore imperative that evidence-based interventions that increase awareness of the LGBTI+ community and inclusivity towards them are developed and implemented in society. Such interventions should be initially aimed at young people themselves, adults working with young people, and healthcare professionals. This will support youth to flourish and live a full and balanced life, irrespective of their sexual orientation or gender identity. It is also important to increase opportunities for young people to contribute to society through volunteering, thus increasing the sense of citizenship and social agency in young people, as well as their sense of cohesiveness.

Ireland is one of the leading countries with regard to having a clear focus on children’s well-being, and its commitment to children and youth is demonstrated by the appointment of a Minister for Child and Youth Affairs, as well as the development of the Better Outcomes, Brighter Futures national youth policy framework and indicator set [21,22]. Ireland was the first country in the world to develop a dedicated national strategy for LGBTI+ youth [39]. The European Union’s LGBTIQ Equality Strategy also highlights the importance of combating inequalities in educational settings [94]. Nevertheless, there is still a need for national and local strategies and policies that are based on the current needs of young people. These should include policies on gender neutral spaces, improving awareness of non-binary gender identities, encouraging inclusion and diversity, and, more specifically, prohibiting discrimination based on gender identity and sexual orientation, in schools and in the workplace. Such policies, and their appropriate implementation, will allow for children of all backgrounds, genders, and sexual orientations to grow and thrive safely in society.

While this study focuses on sexual minority young people, it should be noted that one of the groups reporting perceived discrimination is girls. Discrimination, social exclusion, and inequalities based on gender, as well as the intersectional effects of being a girl and belonging to a minority group [93] are important, especially given the effort that societies make to close the gender gaps in various dimensions and venues. That girls still feel discriminated against is a sign that we must not assume that we have overcome gender discrimination, even if there has been progress at a policy level. Multiple interventions are necessary, most importantly, “gender-proofing” policies and practice developments.

Just as practice and policy need to adapt to accommodate the needs of marginalized young people, so does research. Research often lags behind societal changes, resulting in matters that are deemed important to participants but not to researchers not being included study designs and questionnaires. This highlights the need for child participation in informing research [53], but also emphasizes the need to include underrepresented populations [14] when setting the research agenda. For example, sexual orientation and gender identity are rarely included in youth population studies outside North America, limiting our ability to understand the health disparities in sexual and gender minority young people [95]. Our study clearly demonstrates the difficulties in examining intersectionality, even in a nationally representative youth population study. To allow for better understanding of the impact of intersectionality, researchers should consider oversampling minority groups where possible, or to conduct more targeted research using similar research instruments to allow for comparison with the general population. Temporal changes in sexual and gender minority youths’ health (the effect of the chronosystem) should be investigated with studies using longitudinal designs or repeated measurement at subsequent points.

Rasmussen (2006, p. 2) has pointed out that data on depression, mental ill-health, and suicide are “too often taken as a point of departure when contemplating the lives of LGBT identified young people” [29]. According to Marshall (2010, p. 70), discourses that reproduce sexual minority youth as “always-already victims” have been critiqued as “essentialist presumptions” that socially construct LGB lives as inevitably problematic, disempowered, and traumatized [96]. For instance, mental ill-health and suicidal distress in sexual and gender minority youth can be motivated by a complex web of factors and experiences of which LGB identity is but one small piece of the puzzle [28,88]. We believe that shifting research from the predominant at-riskness and victimizing narrative [27] to a resource- and resilience-oriented agenda may provide clearer direction to those interested in improving sexual and gender minority youths’ lives and well-being. Such a shift is also coherent with the “after-queer” approach in LGBTI+ research [30].

## 5. Conclusions

Adolescents in Ireland, in general, very rarely experience discrimination on any grounds, and perceived discrimination is not strongly affected by gender or social class. A notable exception is that girls were more likely than boys to feel they have been discriminated based on their age and gender. Sexual minority (same- and both-gender attracted) youth perceive more discrimination based on their sexual orientation than their non-minority peers (those who were exclusively attracted to opposite-gender partners or did not report romantic attraction). In addition, those who are attracted to both gender partners reported that they experience discrimination based on their gender and age more often than the other three attraction groups. These findings imply that girls and sexual minority, especially both-gender attracted or bisexual, youth are more vulnerable to stigmatization on multiple grounds, which may be attributed to a negative halo effect.

Additionally, of note is the finding that only around 60 percent of young people report high peer and family support, and positive responses on other indicators of the Better Outcomes, Brighter Futures Outcome 5 (“Connected, respected, and contributing to their world”) are even less frequent. Only around a fifth of young people agreed very much that they made a positive contribution or had freedom in their lives, and around one in ten reported very much taking part in volunteer work. Given that these features contribute to health and well-being, it would be desirable that practice and policy support initiatives that facilitate social skills, connectedness, and civic engagement. In general, these indicators were not strongly associated with gender or social class. Girls were, however, substantially more likely than boys to report high peer support.

Both-gender attracted and not attracted young people were less likely than their opposite- or same-gender attracted peers to report that they feel comfortable being themselves while being with their friends, and not attracted youth were also less likely than the other three groups to report high peer support. Both-gender attracted adolescents were around half as likely to feel valued and respected compared to their non-minority peers. These findings suggest that even within sexual minority young people, psycho-social determinants of health may work differently. Our results are in line with other findings in the literature that young people attracted to both genders or identifying as bisexual are disproportionately affected by stigma, discrimination, social exclusion, and their negative health consequences than their exclusively same- or opposite-gender attracted (lesbian, gay, or heterosexual) peers. Interventions and policies should tackle not only heterosexism and homophobia but monosexism and biphobia as well.

Same-gender attracted young people were almost twice as likely as their opposite-gender attracted peers to report that they take part in volunteer work. Although we do not have data on what type of volunteer work they were engaged in, this finding may indicate that sexual minority youth are more perceptive of social injustice than their non-minority peers and are more motivated to partake in activism and community work. The next iterations of the *LGBTI + National Youth Strategy* of Ireland and similar governmental or state-level initiatives should facilitate civic engagement and volunteering in sexual minority youth as well as an awareness of the specific needs of both-gender attracted or bisexual youth. We believe that further studies in this area have the potential to balance the predominant victimizing narrative on sexual minority young people into a more positive, “after-queer” discourse.

## Figures and Tables

**Figure 1 ijerph-18-01118-f001:**
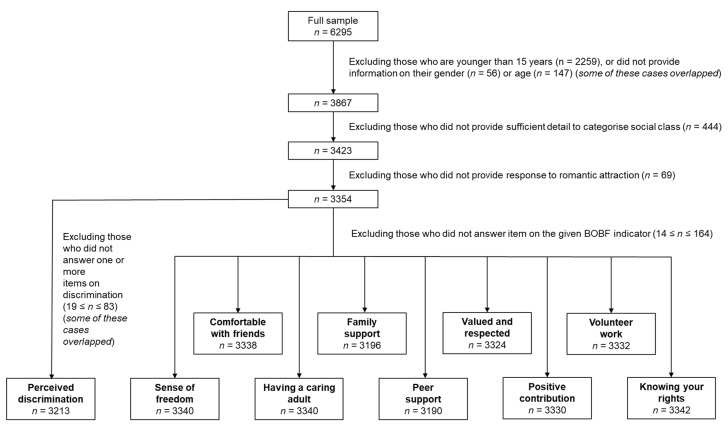
Selection flowchart.

**Figure 2 ijerph-18-01118-f002:**
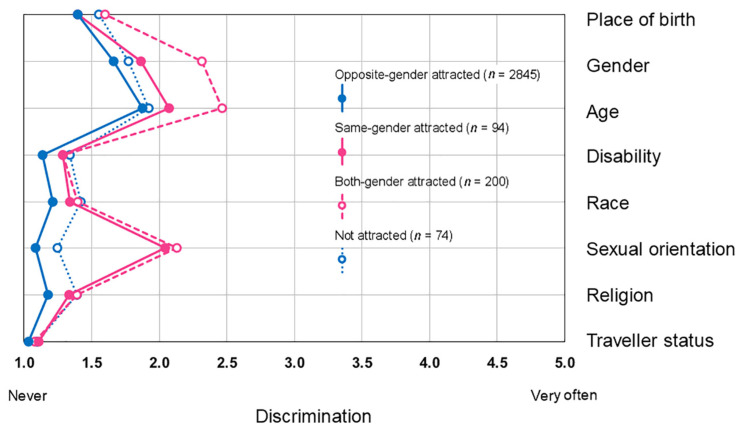
Discrimination profiles in opposite-gender attracted, same-gender attracted, both-gender attracted, and not attracted young people, univariate models (*n* = 3213).

**Table 1 ijerph-18-01118-t001:** Better Outcomes, Brighter Futures (BOBF) framework Outcome 5: “Connected, respected, and contributing to their own world”—Aims and indicator areas within the framework and indicators/response options within the Health Behaviour in School-aged Children (HBSC) Ireland study.

BOBF: Aims under Outcome 5	BOBF: Indicator Areas	Data Sources	HBSC Indicators (Items and Response Options)	HBSC Indicators (Variables and Coding)
Aim 5.1: Sense of own identity, free from discrimination	Discrimination and stigmatization	Central Statistics Office—Quarterly National Household Survey, Special Module on Equality (18- to 24-year-olds) HBSC (13- to 17-year-olds)	How often are you treated unfairly or negatively... ^a^ because of where you, your parents, or grandparents were born? because you are a boy or a girl? because of your age? because of your disability? because of your race? because of your sexual orientation? because of your religion? because you are member of the Traveller community? because of (please write it here): (textbox provided) Never/Hardly ever/Sometimes/Often/Very often	Continuous variables/Discrimination profiles: birthplace gender age disability race sexual orientation religion Traveller status (Never = 1, Hardly ever = 2, Sometimes = 3, Often = 4, Very often = 5)
	Experience of sense of freedom	HBSC (12- to 17-year-olds)	In general, do you feel you have freedom in your life? ^a^ 1 (Not at all)/2/3/4 (Very much)	Dichotomous variable (1 = Very much, 0 = Less than very much)
	Peer acceptance and respect	Growing Up in Ireland (9-year-olds) HBSC (12- to 17-year-olds)	Do you feel comfortable being yourself while with your friends? ^b^ Always/Often/Sometimes/Never	Dichotomous variable (1 = Always, 0 = Less often than always)
Aim 5.2: Part of positive networks of friends, family, and community	Having at least one caring and consistent adult to confide in	Growing Up in Ireland (9-year-olds) EU Survey on Income and Living Conditions Ad hoc Module (18- to 24-year-olds) HBSC (12- to 17-year-olds)	In general, do you have a caring adult you can tell anything to? ^a^ 1 (Not at all)/2/3/4 (Very much)	Dichotomous variable (1 = Very much, 0 = Less than very much)
	Positive parent and family relationships	Programme for International Student Assessment (PISA) (15-year-olds) HBSC (12- to 17-year-olds)	MSPSS, Family subscale ^c^ My family really tries to help me I get the emotional help and support I need from my family I can talk about my problems with my family My family is willing to help me make decisions 1 (Very strongly disagree)/2/3/4/5/6/7 (Very strongly agree)	Scores ≥ 5.5 indicate high family support [42] (1 = High family support, 0 = Low family support)
	Positive relationships with peers	EU Survey on Income and Living Conditions Ad hoc Module (16- to 24-year-olds) HBSC (12- to 17-year-olds)	MSPSS, Friends subscale ^c^ My friends really try to help me I can count on my friends when things go wrong I have friends with whom I can share my joys and sorrows I can talk about my problems with my friends 1 (Very strongly disagree)/2/3/4/5/6/7 (Very strongly agree)	Scores ≥ 5.5 indicate high peer support [42] (1 = High peer support, 0 = Low peer support)
	Perceptions of being valued and respected	HBSC (12- to 17-year-olds)	In general, do you feel you are valued and respected? ^a^ 1 (Not at all)/2/3/4 (Very much)	Dichotomous variable (1 = Very much, 0 = Less than very much)
Aim 5.3: Civically engaged, socially and environmentally conscious	Belief in being able to make a positive contribution to the world	HBSC (12- to 17-year-olds)	In general, do you feel that you make a positive contribution to the world? ^a^ 1 (Not at all)/2/3/4 (Very much)	Dichotomous variable (1 = Very much, 0 = Less than very much)
	Volunteering and altruism	Central Statistics Office—Quarterly National Household Survey, Special Module on Volunteering (18- to 24-year-olds) HBSC (12- to 17-year-olds)	In general, do you take part in volunteer work? ^a^ 1 (Not at all)/2/3/4 (Very much)	Dichotomous variable (1 = Very much, 0 = Less than very much)
	18–24-year-olds who vote in local, regional, national, or European elections and referenda	Central Statistics Office—Quarterly National Household Survey, Special Module on Voter Participation (18- to 24-year-olds)	–	–
Aim 5.4: Aware of rights, responsible and respectful of the law	Children and young people’ awareness of their own rights	HBSC (12- to 17-year-olds)	In general, do you know your rights as a young person? ^a^ 1 (Not at all)/2/3/4 (Very much)	Dichotomous variable (1 = Very much, 0 = Less than very much)
	Respect for laws and the judicial process	(Currently no data collected)	–	–
	Perception of fairness of the law	EU Survey on Income and Living Conditions Ad hoc Module (16- to 24-year-olds)	–	–

Note: ^a^ Developed by the HBSC Ireland team to address gaps in the BOBF indicator set. ^b^ Developed by children as part of a consultation process with Comhairle na nÓg. ^c^ Adapted from the Multidimensional Scale of Perceived Social Support (MSPSS) [43].

**Table 2 ijerph-18-01118-t002:** Descriptive statistics for discrimination, by gender and social class (*n* = 3213).

Sociodemographic Variables	Discrimination Based on							
	Birth-Place ^a^	Gender	Age	Disability	Race	Sexual Orientation	Religion	Traveller Status
	M (SD)	M (SD)	M (SD)	M (SD)	M (SD)	M (SD)	M (SD)	M (SD)
Gender								
Boy (*n* = 1446)	**1.45 (0.86)**	**1.34 (0.75)**	**1.62 (0.96)**	1.17 (0.61)	1.24 (0.73)	1.17 (0.65)	1.19 (0.65)	**1.06 (0.37)**
Girl (*n* = 1767)	**1.38 (0.82)**	**2.01 (1.08)**	**2.16 (1.15)**	1.14 (0.53)	1.23 (0.71)	1.19 (0.64)	1.20 (0.64)	**1.02 (0.25)**
Difference *p*	**0.024**	**<0.001**	**<0.001**	0.076	0.674	0.521	0.600	**0.006**
Difference *r*_ES_ ^b^	**0.040**	**0.336**	**0.246**	< 0.001	<0.001	<0.001	<0.001	**0.048**
Social class								
High (*n* = 1797)	**1.37 (0.79)**	1.73 (1.00)	1.93 (1.10)	1.14 (0.55)	1.23 (0.71)	**1.15 (0.57)**	1.21 (0.66)	**1.03 (0.25)**
Med. (*n* = 1103)	**1.44 (0.87)**	1.70 (1.02)	1.93 (1.13)	1.17 (0.60)	1.22 (0.71)	**1.23 (0.74)**	1.18 (0.62)	**1.05 (0.37)**
Low (*n* = 313)	**1.53 (0.97)**	1.63 (0.97)	1.84 (1.03)	1.15 (0.54)	1.28 (0.80)	**1.17 (0.69)**	1.20 (0.63)	**1.06 (0.40)**
Difference *p*	**0.002**	0.265	0.379	0.423	0.345	**0.004**	0.518	**0.027**
Difference *ω*^2 c^	**0.003**	<0.001	<0.001	<0.001	<0.001	**0.003**	<0.001	**0.002**

Note: ^a^ Discrimination based on the birthplace of the respondent, their parents, or their grandparents. ^b^ Effect size *r*. ^c^ Omega-squared effect size. Med. = Medium social class. Cells highlighted in boldface indicate a statistically significant difference across genders or social classes with respect to perceived discrimination.

**Table 3 ijerph-18-01118-t003:** Descriptive statistics for positive BOBF Outcome 5 indicators in the overall sample, by gender, social class, and romantic attraction.

Sociodemographic Variables and Romantic Attraction	Freedom in Your Life, % (*n*)	Comfortable with Friends, % (*n*)	Having a Caring Adult, % (*n*)	High Family Support, % (*n*)	High Peer Support, % (*n*)	Feeling Valued and Respected, % (*n*)	Making a Positive Contribution, % (*n*)	Taking Part in Volunteering, % (*n*)	Knowing Your Rights, % (*n*)
Overall sample									
Total *n*	3340	3338	3340	3196	3190	3324	3330	3332	3342
Positive answer	23.1% (772)	67.3% (2246)	56.7% (1895)	56.4% (1802)	59.3% (1892)	36.2% (1203)	21.2% (706)	11.5% (384)	26.3% (880)
Gender									
Boy	24.5% (367)	68.5% (1023)	**54.1% (810)**	57.7% (820)	**50.0% (703)**	36.0% (537)	**23.6% (352)**	**10.0% (149)**	26.6% (399)
Girl	22.0% (405)	66.3% (1223)	**58.9% (1085)**	55.3% (982)	**66.7% (1189)**	36.3% (666)	**19.2% (354)**	**12.8% (235)**	26.1% (481)
Association *p*	0.094	0.172	**0.005**	0.177	**<0.001**	0.870	**0.002**	**0.011**	0.719
Association *V* ^a^	0.029	0.024	**0.048**	0.024	**0.169**	0.003	**0.053**	**0.044**	0.006
Social class									
High	23.4% (437)	**65.8% (1231)**	56.7% (1062)	**59.0% (1059)**	59.8% (1066)	36.9% (688)	21.9% (409)	11.9% (223)	25.8% (483)
Medium	21.8% (250)	**71.1% (815)**	55.9% (639)	**53.2% (583)**	59.4% (657)	35.5% (404)	20.4% (233)	11.3% (129)	27.6% (316)
Low	26.3% (85)	**62.7% (200)**	60.1% (194)	**52.5% (160)**	56.3% (169)	34.4% (111)	19.9% (64)	9.9% (32)	25.2% (81)
Association *p*	0.222	**0.002**	0.403	**0.004**	0.535	0.578	0.532	0.562	0.474
Association *V*	0.030	**0.061**	0.023	**0.059**	0.020	0.018	0.019	0.019	0.021
Romantic attraction									
Opposite-gender attracted	23.7% (699)	**68.5% (2020)**	**58.0% (171)**	**57.8% (1634)**	**60.0% (1689)**	**37.6% (1105)**	21.5% (632)	**11.1% (326)**	26.1% (771)
Same-gender attracted	17.2% (17)	**65.0% (65)**	**49.0% (49)**	**48.9% (46)**	**60.8% (59)**	**30.3% (30)**	23.2% (23)	**21.0% (21)**	35.0% (35)
Both-gender attracted	18.6% (39)	**56.3% (117)**	**43.5% (91)**	**42.6% (86)**	**54.0% (109)**	**20.5% (43)**	14.9% (31)	**13.9% (29)**	26.2% (55)
Not attracted	21.3% (17)	**55.7% (44)**	**53.8% (43)**	**48.0% (36)**	**44.9% (35)**	**31.3% (25)**	25.0% (20)	**10.0% (8)**	23.8% (19)
Association *p*	0.164	**<0.001**	**<0.001**	**<0.001**	**0.020**	**<0.001**	0.111	**0.013**	0.239
Association *V*	0.039	**0.074**	**0.076**	**0.084**	**0.056**	**0.091**	0.042	**0.057**	0.036

Note: ^a^ Cramér’s *V* effect size. Cells highlighted in boldface indicate a statistically significant association with gender, social class, or romantic attraction.

**Table 4 ijerph-18-01118-t004:** Descriptive statistics and univariate analysis of variance of discrimination, across romantic attraction groups: opposite-gender attracted (OGA) (*n* = 2845); same-gender attracted (SGA) (*n* = 94); both-gender attracted (BGA) (*n* = 200), and not attracted (NA) (*n* = 74).

Means, Medians and Comparison Across Romantic Attraction	Discrimination Based on…							
	Birthplace ^a^	Gender	Age	Disability	Race	Sexual Orientation	Religion	Traveller Status
Romantic attraction: Means	M (SE)	M (SE)	M (SE)	M (SE)	M (SE)	M (SE)	M (SE)	M (SE)
Opposite-gender attracted	1.39 (0.02)	1.66 (0.02)	1.88 (0.02)	1.13 (0.01)	1.21 (0.01)	1.08 (0.01)	1.18 (0.01)	1.03 (0.01)
Same-gender attracted	1.39 (0.09)	1.86 (0.10)	2.07 (0.11)	1.29 (0.06)	1.34 (0.07)	2.04 (0.06)	1.33 (0.07)	1.11 (0.03)
Both-gender attracted	1.60 (0.06)	2.32 (0.07)	2.47 (0.08)	1.29 (0.04)	1.40 (0.05)	2.13 (0.04)	1.39 (0.05)	1.08 (0.02)
Not attracted	1.55 (0.10)	1.77 (0.12)	1.92 (0.13)	1.34 (0.07)	1.42 (0.08)	1.24 (0.07)	1.39 (0.07)	1.10 (0.04)
Romantic attraction: Medians	Median	Median	Median	Median	Median	Median	Median	Median
Opposite-gender attracted	1	1	1	1	1	1	1	1
Same-gender attracted	1	1	2	1	1	1	1	1
Both-gender attracted	1	2	2	1	1	2	1	1
Not attracted	1	1	1	1	1	1	1	1
Comparison of means								
Kruskal–Wallis *χ*^2^ (*df* = 3)	16.776	72.511	44.975	18.829	18.821	705.610	39.146	2.655
Kruskal–Wallis *p*	0.001	<0.001	<0.001	<0.001	<0.001	<0.001	<0.001	0.448
Parametric *F* (*df* = 3)	4.564	28.048	18.805	9.056	6.724	281.405	10.679	3.877
Parametric *p*	0.003	<0.001	<0.001	<0.001	<0.001	<0.001	<0.001	0.009
Effect size *ω*^2^	0.003	0.025	0.016	0.007	0.005	0.207	0.009	0.003
Statistical power	0.888	~1.000	~1.000	0.996	0.976	~1.000	0.999	0.827
Pairwise comparison (*p* < 0.05)	BGA > OGA	BGA > OGA, BGA > SGA, BGA > NA	BGA > OGA, BGA > SGA, BGA > NA	BGA > OGA, NA > OGA	BGA > OGA	SGA > OGA, BGA > OGA, SGA > NA, BGA > NA	BGA > OGA, NA > OGA	–

Note: ^a^ Discrimination based on the birthplace of the respondent, their parents, or their grandparents.

**Table 5 ijerph-18-01118-t005:** Relative odds for same-gender attracted, both-gender attracted, and not attracted young people, compared to their opposite-gender attracted peers, to show positive outcomes on positive BOBF Outcome 5 indicators.

Positive BOBF Indicators	*n*	OR	*p*	95% CI
Freedom in your life	3340			
Opposite-gender attracted	2951	1 ^a^		
Same-gender attracted	99	0.73	0.150	0.47–1.12
Both-gender attracted	210	0.78	0.101	0.59–1.05
Not attracted	80	0.90	0.618	0.59–1.38
Comfortable while being with friends	3338			
Opposite-gender attracted	2951	1		
Same-gender attracted	100	0.95	0.487	0.82–1.10
Both-gender attracted	208	**0.82**	0.002	0.73–0.93
Not attracted	79	0.81	0.041	0.67–0.99
Having a caring adult	3340			
Opposite-gender attracted	2951	1		
Same-gender attracted	100	0.85	0.102	0.69–1.03
Both-gender attracted	209	**0.75**	<0.001	0.64–0.88
Not attracted	80	0.93	0.467	0.75–1.14
High family support	3196			
Opposite-gender attracted	2825	1		
Same-gender attracted	94	0.85	0.117	0.69–1.04
Both-gender attracted	202	**0.74**	<0.001	0.63–0.87
Not attracted	75	0.83	0.124	0.65–1.05
High peer support	3190			
Opposite-gender attracted	2813	1		
Same-gender attracted	97	1.01	0.876	0.86–1.19
Both-gender attracted	202	0.90	0.110	0.79–1.02
Not attracted	78	**0.75**	0.021	0.58–0.96
Feeling valued and respected	3324			
Opposite-gender attracted	2935	1		
Same-gender attracted	99	0.81	0.159	0.60–1.09
Both-gender attracted	210	**0.54**	<0.001	0.42–0.71
Not attracted	80	0.83	0.266	0.60–1.15
Making a positive contribution	3330			
Opposite-gender attracted	2943	1 ^a^		
Same-gender attracted	99	1.08	0.672	0.75–1.56
Both-gender attracted	208	**0.69**	0.031	0.50–0.97
Not attracted	80	1.16	0.440	0.79–1.71
Taking part in volunteer work	3332			
Opposite-gender attracted	2943	1		
Same-gender attracted	100	**1.90**	0.001	1.28–2.81
Both-gender attracted	209	1.25	0.211	0.88–1.78
Not attracted	80	0.90	0.763	0.46–1.76
Knowing your rights	3342			
Opposite-gender attracted	2952	1 ^a^		
Same-gender attracted	100	**1.34**	0.036	1.02–1.76
Both-gender attracted	210	1.00	0.982	0.79–1.27
Not attracted	80	0.91	0.639	0.61–1.35

Note: ^a^ The overall model was not statistically significant. Odds ratios highlighted in boldface indicate that there is a statistically significant difference in the odds of adolescents in the given romantic attraction group compared to their opposite-gender attracted peers.

## Data Availability

The datasets analyzed in this study can be accessed via the webpage of the HBSC Ireland research team, in accordance with the HBSC data access policy: http://www.nuigalway.ie/hbsc/dataaccess/.

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
