# Peer review of "Connected, Respected and Contributing to Their World: The Case of Sexual Minority and Non-Minority Young People in Ireland"

_ijerph, 2021, doi:10.3390/ijerph18031118_

Round 1
Reviewer 1 Report
The authors have been very conscientious and systematic in their preparation, writing and reporting the results about the survey HBSC. There were smaller issues, where I thought the paper could be improved:
Introduction
- Almost half of the references were published more than five years ago. The subject matter of this paper should be compared with current results. The perception of life in our young people is highly changeable, especially in gender and sexual minorities.
- The authors reflect not that they will make hypotheses, but finally, if they report them. Furthermore, why do they arrive at these "anticipations" as they call them? Relying on the literature could give consistency to the hypotheses.
Method
- More information is needed about the centres and the sample collection procedure. How were the centres chosen? Was it random or by convenience? How were they contacted? How many centres were part of the study out of the total number of centres that could participate? How many centres performed passive and active, informed consent? This may have influenced the results obtained among the centres.
- On what date were the questionnaires administered?
- The authors report that the study adhered to the international study HBSC protocol. It would be useful for readers to briefly explain the criteria that were followed.
- The authors report that they asked the committee for approval, what report number was approved?
Results
- The results are duplicated in text and table. The authors may wish to reduce the text, reflecting the highlights in a summarized manner.
- Grouping the results would help the reading. For example, Both-gender attracted youth were less likely to feel...., to report having a caring adult.
Discussion
- The discussion lacks depth, mainly compared and contrasted with other studies. For example, the authors conclude that females were more likely to feel that they had been discriminated based on their gender or their age. On the other hand, girls were significantly more likely to report high peer support levels than boys. It would be interesting to address what are the implications or recommendations suggested by these findings. There are more examples like this which should be elaborated on.
- The authors conclude "Remarkably, no such difference was found for exclusively same-gender attracted youth". Is it possible that this group is accepted by young people because there has been a change and acceptance towards people's sexual orientation in the younger generations?
- Many of the issues addressed in the involvement section would be appropriate for discussion.
Author Response
Connected, respected, and contributing to their world: The case of sexual minority and non-minority young people in Ireland – Responses to peer reviews
Reviewer 1
The authors have been very conscientious and systematic in their preparation, writing and reporting the results about the survey HBSC. There were smaller issues, where I thought the paper could be improved:
Dear Reviewer,
We would like to thank you for your valuable comments. All your suggestions and queries are addressed below.
Introduction
Almost half of the references were published more than five years ago. The subject matter of this paper should be compared with current results. The perception of life in our young people is highly changeable, especially in gender and sexual minorities.
We absolutely agree with your concern that many references are older than five years. One of the reasons why we felt exploring this area is needed was indeed the scarcity and obsolescence of the available evidence. However, we have changed the following references to more recent ones:
WHO (2010) à Keleher and MacDougall (2016)
Almeida et al. (2009) à Burk, Park and Saewyc (2018)
Cochran, Sullivan and Mays (2003) à Birkett, Newcomb and Mustanski (2015)
The authors reflect not that they will make hypotheses, but finally, if they report them. Furthermore, why do they arrive at these "anticipations" as they call them? Relying on the literature could give consistency to the hypotheses.
We have noted in the previous version of the manuscript that we have not set any specific hypotheses, but we anticipated that age, gender and socio-economic status will impact some of the BOBF indicators (Section 1.2, last paragraph) and that sexual minority youth will be more likely to feel discriminated, less likely to report high levels of social support and they will be more likely to be involved in volunteer work (Section 1.3, last paragraph). In the Discussion, we referred to these anticipations. However, in the updated version we have made changes to outline our hypotheses:
Section 1.2: “We hypothesize that age, gender, and socio-economic status will impact some of the BOBF indicators among 15- to 17-year-old youth in Ireland.”
Section 1.3: “We hypothesize that compared to their non-minority peers, sexual minority youth will be more likely to feel discriminated based on their sexual orientation and other grounds; that they will be less likely to report high levels of social support (from families and peers); and that they will be more likely to be involved in volunteer work.”
We have also reflected on the hypotheses in the Discussion.
Method
More information is needed about the centres and the sample collection procedure. How were the centres chosen? Was it random or by convenience? How were they contacted? How many centres were part of the study out of the total number of centres that could participate? How many centres performed passive and active, informed consent? This may have influenced the results obtained among the centres.
On what date were the questionnaires administered?
The authors report that the study adhered to the international study HBSC protocol. It would be useful for readers to briefly explain the criteria that were followed.
The authors report that they asked the committee for approval, what report number was approved?
We have addressed your above queries as follows:
Section 2.1 (Procedure): “HBSC is a cross-sectional epidemiological study conducted every four years with nationally representative samples of children and adolescents. In line with the international HBSC Protocol [42], a two-stage cluster sampling was conducted to ensure national representativity. First, a proportional sample of schools were randomly selected from the eight geographical regions of Ireland; subsequently, class groups within the participating schools were also randomly selected [47]. School principals were contacted by post; 63% of the invited schools agreed to participate. Data collection took place between April–September 2018. In the present study, responses from an average of 11 young people from 297 classes were analyzed. The study instrument was a paper-based questionnaire that participating youth filled in during school hours. The study was carried out in adherence to the international HBSC study protocol [42] and was approved by the Research Ethics Committee of the National University of Ireland Galway under Decision Ref. REC17-Nov-13. Informed consent was obtained from all participating youth as well as their parents/guardians and school principals. It was at the discretion at the school principals and boards whether active or passive consent from parents was required. Evidence shows that reported health outcomes in young people are independent from the type of parental consent [48], therefore we have not used it as a control variable.”
Results
The results are duplicated in text and table. The authors may wish to reduce the text, reflecting the highlights in a summarized manner.
We have removed repetition of numeric findings throughout Section 3 (Results).
Grouping the results would help the reading. For example, Both-gender attracted youth were less likely to feel...., to report having a caring adult.
We have restructured Section 3.4 (Comparing sexual minority and non-minority youth: Positive BOBF Outcome 5 variables) in this regard.
Discussion
The discussion lacks depth, mainly compared and contrasted with other studies. For example, the authors conclude that females were more likely to feel that they had been discriminated based on their gender or their age. On the other hand, girls were significantly more likely to report high peer support levels than boys. It would be interesting to address what are the implications or recommendations suggested by these findings. There are more examples like this which should be elaborated on.
Thank you for noting these potential implications. Based on your advice we have moved some text from the Introduction to the Discussion and added the following:
Section 4.1 (Descriptive findings): “These results indirectly suggest that interpersonal solidarity among minority girls may, to some extent, counterbalance the negative effects (e.g., stress and anxiety) stemming from discrimination [51]. This may be an important asset in interventions that aim to improve the lives of sexual minority girls.”
Section 4.1. (Descriptive findings): “Another mechanism may be that being connected to communities can improve confidence and self-esteem in young people [56]. A key element of psychotherapeutic interventions for gay and bisexual men is to support them in considering how local LGBTI+ communities may help them in reducing stress and anxiety [57]. Facilitating joining such communities and/or being engaged in volunteering may be particularly beneficial for bisexual or both-gender attracted young people; however, it should be noted that a barrier for them belonging to LGBTI+ communities may be the anticipation that they will be discriminated against based on their bisexuality [58]. More explorative studies are required to better understand the specific needs and experiences of bisexual youth in communities and with voluntary work.”
The authors conclude "Remarkably, no such difference was found for exclusively same-gender attracted youth". Is it possible that this group is accepted by young people because there has been a change and acceptance towards people's sexual orientation in the younger generations?
Unfortunately, from our data we cannot make inferences about such changes as we have used a cross-sectional design. However, we reflected on your comment and suggest that growing tolerance and acceptance may benefit lesbian and gay individuals more than those who identify as bisexual:
Section 4.2. (Descriptive findings): “While from our data we cannot infer the reason for these disparities, it is worth noting that similar to many other Western countries, there is a growing acceptance towards LGBT individuals and issues [81]; however, it seems that bisexual people do not benefit from this shift as much as their lesbian and gay peers. While being lesbian or gay can be perceived by others as a stable identity, bisexuality is often seen as “just a phase” and bisexual individuals report frequent discrimination, identity invalidation and erasure [82]. Therefore, it appears that it is not only individuals that need support and empowerment in being more bi-inclusive, but LGBTI+ communities as well.”
Many of the issues addressed in the involvement section would be appropriate for discussion.
We have relocated some text (e.g., on the need of a more balanced approach to LGBTI+ research) to the Discussion.
Reviewer 2 Report
Can the authors clarify if all study participants are students? If it is the case, please be consistent in the way describing the participants, e.g.
Line 15: 15 to 17-year-olds change to 15 to 17-year-old students
Line 16: young people change to students
Will this be a limitation to this study, i.e. will the results for non-school youths be different?
Author Response
Connected, respected, and contributing to their world: The case of sexual minority and non-minority young people in Ireland – Responses to peer reviews
Reviewer 2
Comments and Suggestions for Authors
Dear Reviewer,
We would like to thank you for your valuable comments. Your suggestions and queries are addressed below.
Can the authors clarify if all study participants are students? If it is the case, please be consistent in the way describing the participants, e.g.
Line 15: 15 to 17-year-olds change to 15 to 17-year-old students
Line 16: young people change to students
Given that ‘student’ has a connotation of people attending colleges or high schools, we have eliminated this term as well as ‘children’, and used ‘young people’ or ‘youth’ throughout the Method, Results and Discussion sections.
Will this be a limitation to this study, i.e. will the results for non-school youths be different?
Thank you for raising this very important point. We have added the following text to section 4.3. Limitations and strengths:
“Finally, it should be noted that since HBSC collects data in classrooms, young people who were absent on the day of the data collection or attend youth centers or out of school services will inevitably be excluded from the sample. Given that sexual minority youth tend to miss school due to health problems [77] or due to fear of harassment and bullying [36], they are probably underrepresented in the present study. Further work is needed to include a broader sample of young people both within and outside the traditional school setting (e.g., by using community sampling).”
Reviewer 3 Report
The article concerns the psychosocial factors influencing the health of adolescents who belong to sexual minorities, as compared to their peers from majority groups, on the basis of Outcome 5 of the Irish Better Outcomes, Brighter Futures (BOBF): the national youth policy framework „Connected, respected, and contributing to their world”. The issue presented in the article is significant due to the risk of discrimination and stigmatization of people from marginalized groups and due to the crucial importance of this phenomenon for young people’s health and well-being.
1.The summary is well-written and it contains information crucial for the article; it is also clear and understandable.
2. The introduction is devised in a coherent and consistent way; it outlines well the theoretical and social background as well as it justifies the analyses included in the article. In my opinion, it is worth characterising the situation of sexual minorities in Ireland as early as in the introduction, since it outlines the social context of the results obtained. Information on the issue appear in the article, but only in the final part (4. Implications for practice, policy, and research).
3. The introduction contains some expectations towards the research results. I suggest that they were formulated as specific research hypotheses
4.Procedures and measures have been described clearly and succinctly. The results have been presented in an appropriate, clear and orderly way. They have also been well-interpreted and discussed.
5. The article reads well and it is interesting and cognitively valuable. It also has significant practical applications, for example concerning actions directed at increasing awareness of LGBT community and voluntary work as especially constructive form of social activation of young people.
6. The analyses are conducted according to the most current trends and they emphasize positive aspects of health, positive experiences and factors favourable to high quality life.
Author Response
Connected, respected, and contributing to their world: The case of sexual minority and non-minority young people in Ireland – Responses to peer reviews
Reviewer 3
The article concerns the psychosocial factors influencing the health of adolescents who belong to sexual minorities, as compared to their peers from majority groups, on the basis of Outcome 5 of the Irish Better Outcomes, Brighter Futures (BOBF): the national youth policy framework „Connected, respected, and contributing to their world”. The issue presented in the article is significant due to the risk of discrimination and stigmatization of people from marginalized groups and due to the crucial importance of this phenomenon for young people’s health and well-being.
Dear Reviewer,
We would like to thank you for your positive feedback and valuable comments. Your suggestions and queries are addressed below.
1.The summary is well-written and it contains information crucial for the article; it is also clear and understandable.
Thank you.
- The introduction is devised in a coherent and consistent way; it outlines well the theoretical and social background as well as it justifies the analyses included in the article. In my opinion, it is worth characterising the situation of sexual minorities in Ireland as early as in the introduction, since it outlines the social context of the results obtained. Information on the issue appear in the article, but only in the final part (4. Implications for practice, policy, and research).
We have added to Section 1.3: “While there have been some studies with sexual minority youth in Ireland [36-38], the research landscape is rather bleak and the existing studies concentrate on the negative aspects of belonging to sexual minorities. The National LGBTI+ Youth Strategy 2018-2020 [39] prioritizes conducting more comprehensive and balanced research to better understand the specific needs, and developmental assets, of sexual and gender minority youth. In recent years, Ireland has advanced LGBTI+ equality at a structural level, with the passing of Marriage Equality legislation and the Gender Recognition Act in 2015. However, there is still much work to do in achieving more inclusive environments for LGBTI+. For instance, LGBTI+ young people still report a lack of understanding and acceptance [40], as well as barriers to accessing inclusive and supportive services [41].”
- The introduction contains some expectations towards the research results. I suggest that they were formulated as specific research hypotheses
We have changed “anticipations” mentioned in the introduction to more formal hypotheses:
Section 1.2: “We hypothesize that age, gender, and socio-economic status will impact some of the BOBF indicators among 15- to 17-year-old youth in Ireland.”
Section 1.3: “We hypothesize that compared to their non-minority peers, sexual minority youth will be more likely to feel discriminated based on their sexual orientation and other grounds; that they will be less likely to report high levels of social support (from families and peers); and that they will be more likely to be involved in volunteer work.”
We have reflected on the hypotheses in the Discussion.
- Procedures and measures have been described clearly and succinctly. The results have been presented in an appropriate, clear and orderly way. They have also been well-interpreted and discussed.
Thank you.
- The article reads well and it is interesting and cognitively valuable. It also has significant practical applications, for example concerning actions directed at increasing awareness of LGBT community and voluntary work as especially constructive form of social activation of young people.
Thank you.
- The analyses are conducted according to the most current trends and they emphasize positive aspects of health, positive experiences and factors favourable to high quality life.
Thank you.